

# Saproxylic Diptera assemblages in a temperate deciduous forest: implications for community assembly

Julia J. Mlynarek[1,2], Amélie Grégoire Taillefer[2,3] and Terry A. Wheeler[2,†]

[1] Harrow Research and Development Centre, Agriculture and Agri-Food Canada, Harrow, ON, Canada
[2] Department of Natural Resource Sciences, McGill University, Ste-Anne-de-Bellevue, QC, Canada
[3] BioÉco Environnement, Pincourt, QC, Canada
† Deceased author.

Corresponding author
Julia J. Mlynarek,
julia.mlynarek@gmail.com

## ABSTRACT

Saproxylic insects, those associated directly or indirectly with decaying wood for all or part of their life cycle, compose a large proportion of forest organisms. Flies (Diptera) are often the most abundant and species-rich group of insects in forest microhabitats, yet most work to date on saproxylic insect diversity and ecology has focused on beetles (Coleoptera). We compared saproxylic Diptera assemblages reared from two tree species (sugar maple and American beech) at two stages of decay (early/young and advanced/old) for a total of 20 logs in an eastern Canadian Nearctic old-growth forest. We found that communities are distinct within both species type and decay stage of wood. Early decay stage wood is more variable in community composition than later decay stage; however, as the age of the decaying wood increases, the abundance of Diptera increases significantly. Most indicator species are discernible in later decay stage and wood type. We venture to suggest that stochastic and deterministic processes may play a role in driving Diptera communities in temperate deciduous forests. To retain the highest saproxylic Diptera diversity in a forest, a variety of decaying wood types at different stages of decomposition is necessary.

## INTRODUCTION

A proportion of forest invertebrates are wood- or bark-inhabiting species and have a significant role in wood decomposition in temperate regions (*Ulyshen, 2016*). The reduction of old growth forest area and implementation of forest management practices, such as clear cutting and timber harvesting, negatively impact several wood associated arthropod taxa (*Buddle et al., 2006*; *Pohl, Langor & Spence, 2007*; *Grodsky et al., 2017*). Many studies have demonstrated the importance of coarse woody debris (CWD) in maintaining forest arthropod diversity, especially of saproxylic species associated directly or indirectly with decaying wood for all or part of their life cycle (*Speight, 1989*; *Grove, 2002*).

Saproxylic coleoptera are considered a model system when studying insect diversity and abundance in decaying wood (see *Stokland, 2012* for review). *Irmler, Helier & Warning (1996)* showed that beetle abundance and species richness increased with the age of three tree species (beech, alder and spruce) in European forests. In Canadian boreal forests, species richness of Coleoptera was higher in older logs although abundance was lower (*Hammond, Langor & Spence, 2004*; *Stokland & Siitonen, 2012*). In addition to age of decaying logs, the tree species has been shown to play a role in maintaining saproxylic Coleoptera diversity (*Jonsell, Hansson & Wedmo, 2007*).

Most work to date on saproxylic insect diversity and ecology has focused on beetles but flies (Diptera) are often as or more abundant and species-rich in the same microhabitats (*Schiegg, 2000*; *Rotheray et al., 2001*; *Vanderwel et al., 2006*; *Persson, Lenoir & Vegerfors, 2013*). *Stokland & Siitonen (2012)* stated that there are more saproxylic Diptera than Coleoptera in Nordic countries; additionally, other studies in Europe have found that the Diptera families: Mycetophilidae, Sciaridae and Cecidomyiidae, are particularly dominant and species-rich in decaying logs (*Irmler, Helier & Warning, 1996*; *Økland, 1994*; *Hövemeyer & Schauermann, 2003*). With the exception of work by *Work & Hibbert (2011)*, there has been little empirical study of diversity patterns or microhabitat use in saproxylic Diptera in North America, particularly at taxonomic scales below that of family.

Previous studies of saproxylic Diptera diversity in North America have been based on coarse taxonomic sorting (generic or family-level identification only; *Vanderwel et al., 2006*; *Batzer & Braccia, 2008*; *Dennis et al., 2017*); however, coarse taxonomic sorting of Diptera may not provide the necessary detail to study community assembly in this group. Many Diptera families are ecologically diverse, with documented differences in microhabitat use, trophic role and seasonal activity even within a genus (*Ferrar, 1987*; *Lévesque-Beaudin & Wheeler, 2011*). Species-level identification of Diptera communities, where possible, is necessary to accurately document fine-scale patterns of habitat use and community ecology. If conservation of saproxylic insect fauna is to be added to forest management guidelines, a better understanding of community patterns, with identifications done at the species-level, is needed.

We studied saproxylic Diptera reared from decaying logs of sugar maple (*Acer saccharum* Marshall) and American beech (*Fagus grandifolia* Ehrhart) at two stages of decay (ca. 2 years, and ca. 6 years after death) in an eastern Nearctic old-growth forest. Sugar maple and American beech are the two dominant tree species in this type of forest. The objective of our study was to determine the effect of host-tree species and decay stage on community composition and community assembly of saproxylic Diptera. Based on past research focused on saproxylic Coleoptera (reviewed in *Stokland, 2012*; *Stokland & Siitonen, 2012*), we predicted that if Diptera communities react similarly to wood decay, they should be affected by the decay stage and tree species. We expected that there would be some species turnover between decay stages, and that community composition should be somewhat unique between tree species (*Stokland & Siitonen, 2012*).

## MATERIAL AND METHODS

### Study site and sampling

The study was conducted at the Mont Saint-Hilaire Biosphere Reserve in Southern Quebec, Canada (45°32′40″N, 73°9′5″W) within 500 m of the shore of Lac Hertel at the Reserve between 173 and 223 m elevation (see cantrenature.qc.ca for information). The Reserve is dominated by a closed canopy, hardwood beech-sugar maple forest, which is the most common forest type in Southern Quebec.

We chose sugar maple and American beech because they are the dominant trees at the Reserve and they have similar wood hardness (Janka hardwood index: 1,450 for sugar maple vs. 1,300 for beech), decay rates and other wood properties (*Johnson et al., 2014*). The main difference between the two species is the characteristics of their bark: American beech has smoother bark than sugar maple.

Five fallen logs per tree species (sugar maple and American beech) and per two decay stages (early/young and advanced/old) were selected for a total of 20 decaying logs. The decay stages were determined by visual inspection and based on the seven stages from *McCullough (1948)*, modified to deciduous trees by *Crites & Dale (1998)*. Early decay logs were characterized by having 10–20% of the bark absent, and the first centimeters of the logs had been infiltrated by moisture and sapwood structural decay. Advanced decay logs had 80% or more of the bark absent and the heartwood had been infiltrated with moisture and decay. The logs chosen were in stands dominated by the particular species and in close proximity to other logs of the same species (i.e., the sugar maple logs were in sugar maple dominated stands in proximity of other maple sugar logs) to ensure connectivity and allow saproxylic insect colonization from the appropriate species to the logs (*Schiegg, 2000*). All the logs had to be between 18 and 22 cm in diameter. In May 2004, the selected branchless logs were cut (1.2 m in length) on site. Once cut, each log was repositioned in its original location on a plastic ground sheet (to exclude insects emerging from soil or litter). Each log was then covered by an emergence trap (with a collecting jar at one end), which was placed on and sealed to the plastic ground sheet, to prohibit any further colonization (*Irmler, Helier & Warning, 1996*). The insects emerging from the log were collected into the collecting jar filled with a 50% solution of propylene glycol and water. These jars were collected weekly from June 3 to September 16, 2004 (16 weeks). This period corresponds to the vegetation period and the main period of insect activity.

### Specimen preparation and identification

All insects were stored in 70% ethanol, then chemically dried and mounted for identification. Diptera were identified to named species where taxonomic expertise and available literature permitted (Table S1). Where identification to named species was unavailable specimens were sorted to morphospecies based on standard morphological characters (*McAlpine et al., 1981*) used in the literature for identification of related taxa. The specimens identified to morphospecies were numbered with a unique identifier for this study (e.g., *Sciara* sp. jm1), which are databased. These unique identifiers will allow future research (taxonomic or ecological) easy access for verification and comparison of these specimens to others in

similar studies. All specimens are deposited in the Lyman Entomological Museum (McGill University, Ste-Anne-de-Bellevue, QC, Canada).

Analyses were based on all Diptera except: Cecidomyiidae (focus of a separate study); the sciarid genera *Bradysia*, *Corynoptera*, *Lycoriella* and *Scaptosciara*, each of which was represented by multiple morphospecies that can only be distinguished after dissection and slide-mounting; and the phorid genus *Megaselia*, which was represented by several morphospecies that could not be reliably distinguished.

Each species was assigned to a trophic group based on their larval feeding habit among fungivore, saprophage, predator, parasite, omnivore and phytophage. Trophic habits were determined according to information given in *Ferrar (1987)* and other available literature (*Pritchard, 1983*; *Brown, 1985*; *Brown & Hartop, 2017*).

## Statistical analyses

### Diversity patterns

Using the pooled data for each treatment, Simpson's reciprocal diversity index (1/D) was calculated as an evenness measure with 10 individuals for the upper abundance limit for rare species and 100 runs of randomization for estimators (*Hill, 1973*). Extrapolated species richness was assessed using a bias-corrected Chao index (*O'Hara, 2005*). Species diversity and sampling efficiency was examined using individual- and sample-based rarefaction curves (*Gotelli & Colwell, 2001*). Expected species richness of each treatment was calculated using rarefaction estimates standardised to 445 individuals which is the lowest number of individuals collected in the young maple (YM) treatment. All analyses were performed using the vegan package (*Oksanen et al., 2012*) in R 3.4.2 (*R Development Core Team, 2017*).

The habitat association of each species was examined using indicator species analysis performed with the function multipatt in the package indicspecies (*De Caceres & Legendre, 2009*). Each species was tested for its association with tree species and decay stage separately or in combinations. The significance of species association was assessed with a permutation test using 999 permutations. Only species with $\geq$10 individuals and an indicator value $\geq$45% were considered. We tested whether indicator values differed significantly between feeding guilds using the Kruskal–Wallis test due to the small sample size.

A generalized linear model (GLM) (*McCullagh & Nelder, 1989*) with Poisson distribution for counts was used to evaluate the relationships between abundance, species richness, estimated species richness (Chao) and Simpson's diversity index as response variables with decay stages and tree species as predictors. Each predictor was tested one by one, then in combination. As *Margules, Nicholls & Austin (1987)* demonstrated that interactions between variables often provide a higher predictive power than the same variables separately; the interaction was tested and ranked according to Akaike Information Criterion (AIC). The model with the lowest AIC value was selected as the best model.

### Community composition

Community composition among treatments was compared using non-metric multidimensional scaling (NMDS). Permutational multivariate analysis of variance

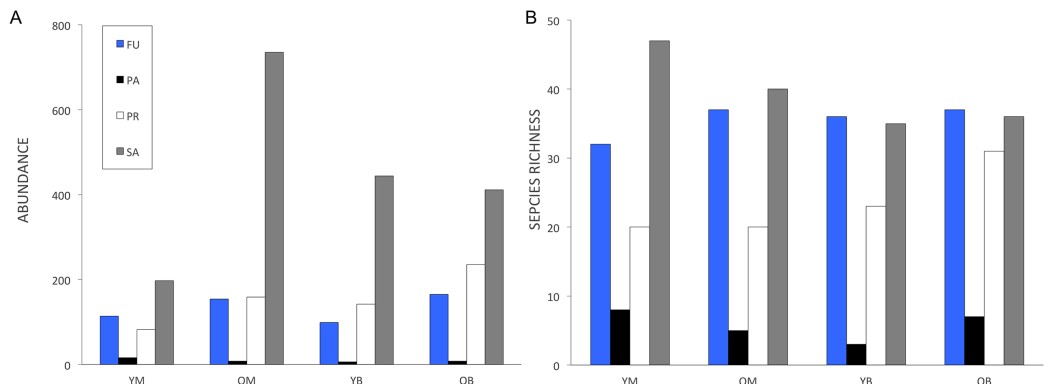

**Figure 1 Decaying wood saproxylic Diptera trophic structure in a temperate deciduous forest in southern Quebec depending on feeding guild for (A) abundance and (B) species richness.** FU, Fungivores; PR, predators; PA, parasites and SA, saprophages; YM, young maple; OM, old maple; YB, young beech and OB, old beech.                

based on a Bray-Curtis distance matrix was used to assess significance of differences among treatments for overall species assemblages using the function Adonis. Prior to NMDS and Adonis, species abundances were Hellinger transformed (*Legendre & Gallagher, 2001*), because this transformation is particularly suited to species abundance data, as it gives low weights to low counts and many zeros.

To test for tree host and age of CWD on Coleoptera species we used null model analysis with EcoSim version 7.71 (*Gotelli & Entsminger, 2010*). This tested whether or not Diptera species collected in the different treatments are distributed in a random manner with regard to each other. Community structure indices were computed for all logs pooled by treatment types and each tree species and decay stage separately. Co-occurrence analysis was performed using the *C*-score (*Stone & Roberts, 1990*) index that measures the average number of checkerboard units (species mutual exclusion) between all possible pairs of species in a presence-absence matrix. To find a non-random pattern of species co-occurrence, the *C*-score should be significantly lower or higher than expected by chance. A Monte Carlo null model simulation was used to randomize the matrix 5,000 times with the sequential swap algorithm and fixed sum rows and columns constraints.

# RESULTS

## Diversity patterns

A total of 3,034 specimens representing 227 named species and morphospecies were used in the analyses (Appendix 1). Overall the most abundant families were Milichiidae (891 specimens), Empididae (470 specimens), Limoniidae (438 specimens) and Mycetophilidae (306 specimens). The most species rich families were Mycetophilidae (50 species), Empididae (29 species), Sciaridae (20 species) and Limoniidae (19 species). Milichiidae abundance was divided between only two species, with *Neophyllomyza quadricornis* Melander being the most abundant species overall (721 specimens; *Brochu & Wheeler, 2009*). Trophic structure was similar in all four treatments (Fig. 1). Saprophages were the most abundant, followed by fungivores or predators and parasites being the

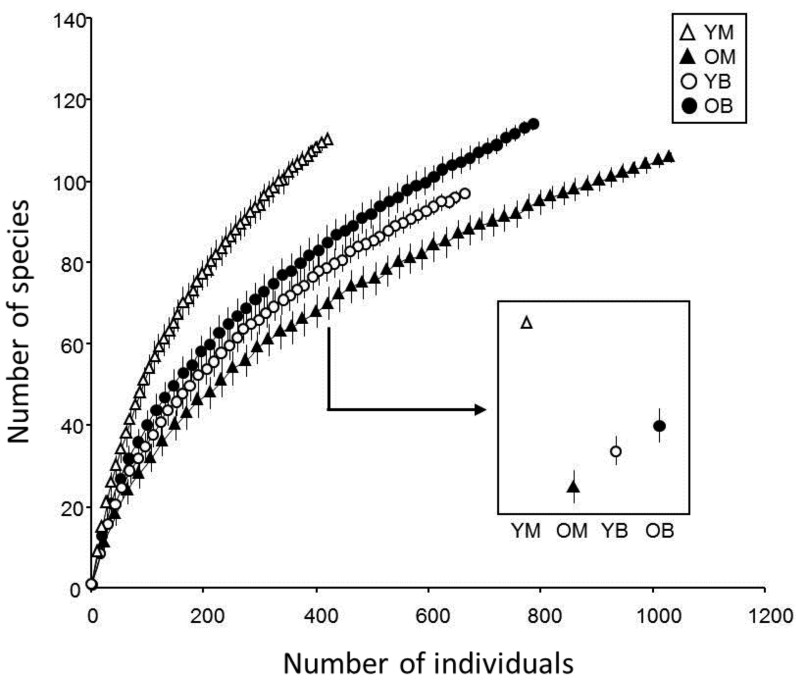

**Figure 2 Rarefaction estimates species richness (±1 SE) of saproxylic Diptera in a Quebec deciduous forest plotted against number of individuals at different wood decay stages and tree species.** YM, Young maple; OM, old maple; YB, young beech and OB, old beech.

least abundant (Fig. 1A). As for trophic richness, saprophages were the most species rich, followed by fungivores, predators and parasites (Fig. 1B). Phytophages and omnivores were excluded as they were represented by only two species each.

In the treatments, 48% of the collected species were represented by only one or two specimens. Chao indices suggested that between 83 and 88% of the species present were collected. The rarefaction curves for all treatments did not reach an asymptote (Fig. 2) and the sample-based rarefaction curves showed that none of the sampling was distinctly better, as curves for the four treatments were not significantly different (Fig. 3).

Indicator species analysis revealed that 15 species among fungivores, saprophages, predators and parasites were significantly associated with tree species, decay stage or a combination of the two (Table 1); however, there were no indicator species associated with just young decay. All indicator values were relatively low, below 75 and there was no tendency for any of the feeding guilds to have higher indicator value than any other feeding guild (Chi-square = 2.96; d$f$ = 3; $P$ = 0.40).

By just describing the total numbers of specimens collected in each log type, old maple logs (OM) had the most specimens collected followed by old beech (OB), young beech (YB) and YM (Table 2). The diversity (rarefaction estimate) was higher in YM than OM. On beech, diversity did not significantly differ between old and young wood. The Simpson's index was lower in OM and YB than in YM and OB.

The results from the GLM with the lowest AIC value showed that decay stage ($F$ = −2.89; $P$ < 0.01) and tree species ($F$ = 5.91; $P$ < 0.01) had a combined effect on

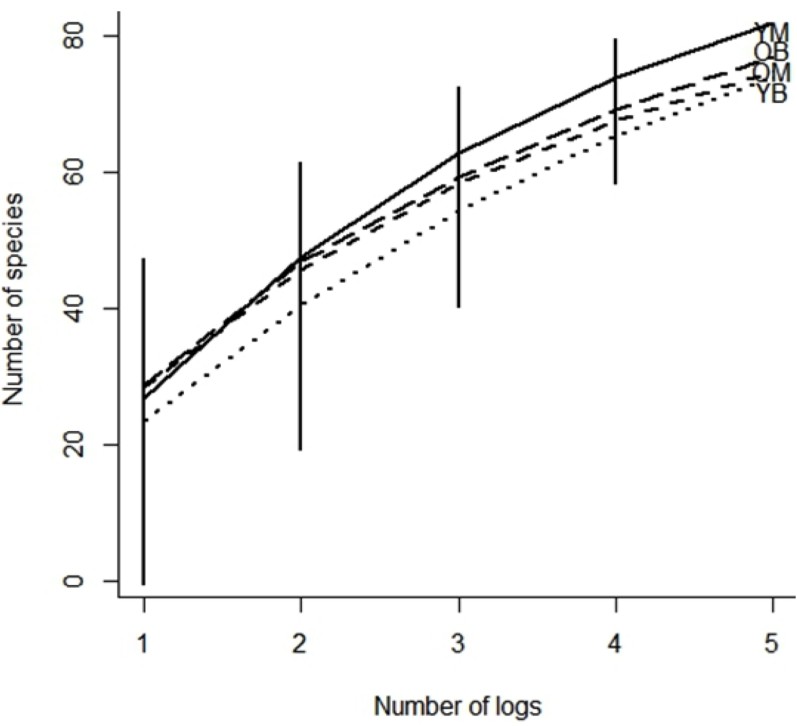

**Figure 3 Species accumulation curve of southern Quebec saproxylic Diptera collected from decaying logs.** (YM, young maple; OM, old maple; YB, young beech and OB, old beech). The bars indicate the 95% confidence interval based on standard deviation.

**Table 1 Southern Quebec decaying wood Saproxylic Diptera indicator species analysis showing species with a significant association (*P* < 0.05) with treatments and an indicator value >45.**

| Treatment | Trophic group | Species | Indicator value | *P*-value |
|---|---|---|---|---|
| Maple | Fungivore | Platosciara sp. jm 2 (Sciaridae) | 57 | 0.036 |
| | Fungivore | Leia sp. jm 1 (Mycetophilidae) | 49 | 0.019 |
| Beech | Predator | Tachypeza sp. jm 3 (Empididae) | 67 | 0.017 |
| Old decay | Fungivore | Discobola annulata (Limoniidae) | 55 | 0.047 |
| | Fungivore | Sciara sp. jm 7 (Sciaridae) | 58 | 0.044 |
| | Predator | Allodromia testacea (Empididae) | 56 | 0.042 |
| | Predator | Tachypeza sp. jm 3 (Empididae) | 58 | 0.046 |
| | Saprophage | Neophyllomyza gaulti (Milichiidae) | 72 | 0.015 |
| Old beech | Fungivore | Sciara sp. jm 7 (Sciaridae) | 45 | 0.047 |
| | Predator | Leptopeza sp. jm 1 (Empididae) | 60 | 0.023 |
| | Predator | Tachypeza sp. jm 3 (Empididae) | 54 | 0.013 |
| Old maple | Saprophage | Neophyllomyza quadricornis (Milichiidae) | 46 | 0.033 |
| | Saprophage | Homoneura philadelphica (Lauxaniidae) | 52 | 0.025 |
| Young beech | Saprophage | Gaurax atripalpus (Chloropidae) | 60 | 0.038 |
| Young maple | Parasite | Allophorocera sp. jm (Tachinidae) | 60 | 0.031 |

species abundance ($F = -9.71$; $P < 0.01$) (Table 3). However, the GLM did not retain any significant model ($P < 0.05$) for species richness, estimated species richness (Chao) and Simpson's diversity index.

**Table 2 Sample size, estimates and diversity indices of saproxylic Diptera in decaying wood in each treatment.**

| Treatment | N | $S_{est}$ | Simpson | Chao |
|---|---|---|---|---|
| Young maple | 445 | 113 ± 0 | 41.5 | 91.6 ± 6.3 |
| Old maple | 1,066 | 72 ± 3.9 | 5.7 | 90.3 ± 9.6 |
| Young beech | 698 | 81 ± 3.2 | 7.3 | 84.6 ± 6.3 |
| Old beech | 825 | 87 ± 3.8 | 21.3 | 92.8 ± 9.4 |

Note:
N, Number of individuals; $S_{est}$, rarefaction estimates of species richness (species ± SE, standardized at 445 individuals), Simpson's diversity index and Chao index.

**Table 3 Summary of generalized linear models (GLM) showing the effect of environmental variables (decay stage and tree species) on saproxylic Diptera abundance in decaying logs in southern Quebec.**

| Formula | AIC | Residual degrees of freedom | Residual deviance | Deviance |
|---|---|---|---|---|
| abun ~ decay | 1216.6 | 18 | 1,082 | 1267.5 |
| abun ~ tree | 1402.1 | 18 | 1267.5 | 1267.5 |
| abun ~ decay * tree | **1124.1** | 16 | 985.5 | 1267.5 |
| abun ~ decay + tree | 1218.6 | 17 | 1,082 | 1267.5 |

Note:
Value in bold highlights the best fit model.

## Community composition

Based on the Adonis, decay stages and tree species had significant effect on community composition. The four treatments were significantly different (Adonis $R^2 = 0.21$, $P = 0.007$) in species assemblages and the NMDS results demonstrate that young decay stages are more variable in community composition than older decay stages (Fig. 4). Therefore, every decay stage and tree species has a distinct community composition, although decay stage and tree species only explain 21% of the variation.

Co-occurrence at the treatment level showed no difference from that expected by chance ($P \geq 0.05$), in maple, beech and early decay stage, although the community in the advanced decay stage exhibited high species segregation ($P_{(observed > expected)} = 0.015$).

## DISCUSSION

As expected, host-tree species and decay stage impact community composition and community assembly of saproxylic Diptera when examined at the species level and community assemblages in both tree species and decay stage are relatively distinct. Therefore, to retain the highest diversity of Diptera in a forest, a variety of decaying wood species at different stages of decomposition is necessary. Decay stage also has a significant effect on abundance of Diptera; as the age of the decaying wood increases so does the abundance of Diptera, and the number of indicator species. These patterns are discernible in advanced decay stage and in the interactions between log type and decay stage. Additionally, there is significant difference of Diptera species at advanced decay stages; the communities are quite unique in each log.

Our results show that high species turnover was present among individual logs. These results are consistent with the other species-specific study of North American

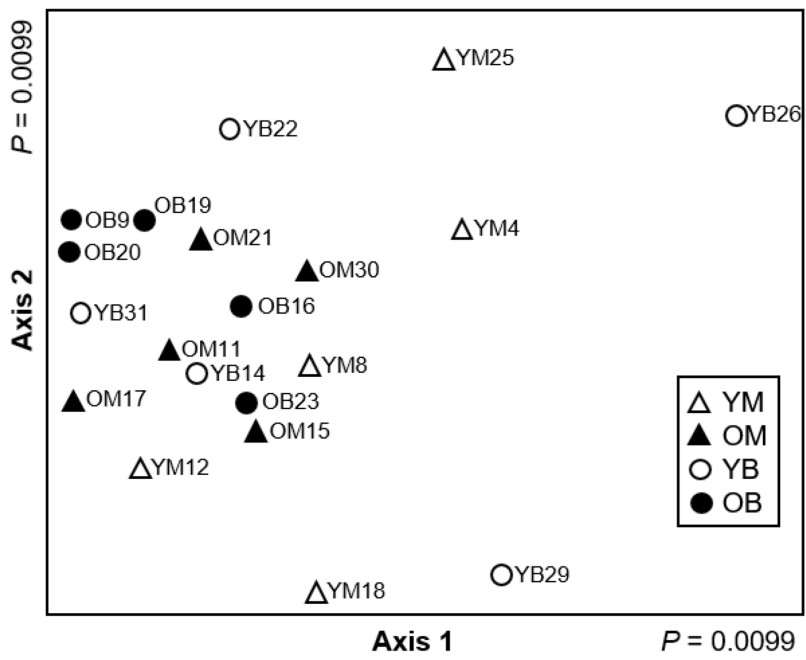

**Figure 4 Non-metric multidimensional scaling ordination (stress = 0.088) based on Hellinger transformed abundance of saproxylic Diptera species in decaying wood in southern Quebec.** The two axes of a two-dimensional solution are plotted. Young maple (YM; open triangles), old maple (OM; closed triangles), young beech (YB; open circles) and old beech (OB; closed circles).

Diptera (*Work & Hibbert, 2011*), which observed compositional similarity between logs as low as 20%. CWD has been shown to be important at the stand level (e.g., landscape scale; *Work et al., 2004*). However, based on our results, we suggest that the scale of the log is essential for microhabitat diversity, which may be due to saproxylic Diptera having relatively poor active dispersal capabilities (*Schiegg, 2000*) and specific microhabitat requirements (*Siitonen, 2012*). The patchiness in assemblage pattern suggests that to develop comprehensive biomonitoring and biodiversity conservation strategies, research must focus on several scales and provide a maximum of decaying tree host species at different stages of decay.

The species of the decaying log is somewhat important in determining the community assembly of Diptera. Our results show that there are seven indicator species found to be specific to a tree species when interacting with decay stage and an additional three were found to be specific to one of the tree species no matter the decay. This is interesting considering that few Diptera species feed directly on wood (*Teskey, 1976*). There must be other characteristics within the tree species that encourage the development of specific species that should be looked at in future studies regarding Diptera communities.

Based on our results, age of CWD has an important influence on the Diptera community assembly. Advanced decay stage logs have a more distinct (10 indicator species no matter the tree species and five specifics to decay and tree species) and less variable community than early decay logs (only two indicator species specific to both decay stage and tree species). This is potentially due to changes in characteristics of the decaying

wood as suggested by *Persson, Lenoir & Vegerfors (2013)*. They concluded that microarthropods community changes as the physical and chemical characteristics of the wood change. Unlike similar studies of Coleoptera (*Jonsell, Hansson & Wedmo, 2007*) and midges (*Irmler, Helier & Warning, 1996*), we did not observe an increase in species richness with age of log decomposition. A potential reason could be because of the differences in age of the decaying logs observed in our study (several years between young logs and old logs) compared to differences in ages of CWD in the other studies. We did, however, see higher Diptera abundance in advanced decay logs, which allows us to conclude that the conditions of older logs can support a higher number of individuals. A potential reason for the higher abundance in old logs could be the increased presence of fungi, nitrogen, and water content and decreased carbon content in aging wood (*Hövemeyer & Schauermann, 2003*; *Palviainen et al., 2010*). Advanced stage decay logs, therefore, provide more readily available food sources and ovipositing resource to saprophagous and fungivorous Diptera (e.g., Milichiidae and Mycetophilidae; *Buxton, 1960*; *Jakovlev, 2011*) than early stage decay logs and can, in turn, support higher abundances of predatory species in families such as the Empididae.

The indicator species from our study represent a variety of feeding guilds (fungivore, saprophage and predator). We can only speculate on specific feeding specialization and dispersal capabilities of each species collected because there is sometimes not enough life history information on these species in the literature. Increased studies of life history in North American Diptera in CWD would, in part, help understand the processes that drive Diptera communities in decaying logs.

Even though it was not our intention at the beginning of the study, the change in community structure made us consider the processes (stochastic vs. deterministic) that could be driving communities depending on decay stage and tree species. Both stochastic and deterministic processes could be occurring; however, the significance of each of the processes depends on the time of observation during the ecological succession. Based on our results, newly fallen logs are likely to be colonized by many Diptera species through random events, because community composition is more variable in young decay stage. As decaying wood ages, the assemblage of the community becomes much more specific and somewhat unique for each log leading us to conclude that community assembly in older logs is driven by deterministic processes associated to decay and not as much to host-tree species. Our NMDS results and the number of indicator species compared between the two stages of wood decay show that later decay stages are dominated by species that are specialised to live on or in older decaying wood.

Due to the temporal pattern apparent in our results, we suggest that future studies could profitably look more deeply into the continuum between stochastic and deterministic community assembly and place their analyses into a niche/neutral model framework. An interesting long-term experiment would be to control the age of logs by following community assembly from the moment a log is cut through its decay to determine at which moment the switch between stochastic and deterministic process occurs. Further studies are also needed to assess the mechanisms driving the distinct communities in old logs: resource utilization, interspecific competition, time (phenology)

partitioning or space (where they are found in or on the logs) partitioning (*Loreau, 1989*; *Gilbert, Srivastava & Kirby, 2008*).

## CONCLUSION

We set out to determine how Diptera communities differ between decay stage and tree species of logs because it is an understudied but diverse group of insects in forest habitats. Using a morphospecies approach we were able to obtain finer scale identification, but to fully understand community assembly patterns, more identification tools (including molecular techniques) of the broader community (including all invertebrates and fungi) and in-depth life-history information are needed. Our results suggest that Diptera community assembly may be driven by both stochastic and deterministic processes which is in support of recent studies in community assembly (*Gravel et al., 2006*; *Thompson & Townsend, 2006*; *Ellwood, Manica & Foster, 2009*; *Barber & Marquis, 2011*; *Ferrenberg, Martinez & Faist, 2016*; *Grégoire Taillefer & Wheeler, 2017*). We have also demonstrated that to retain the highest saproxylic Diptera diversity in a forest, a variety of decaying wood types at different stages of decomposition is necessary.

## ACKNOWLEDGEMENTS

We thank Duncan Selby, whose M.Sc. project on saproxylic gall midge diversity provided the Diptera specimens for this study and Kathrin Sim for reading a draft version. McGill University, Martin Lechowicz and Benoit Hamel provided access and logistic support at McGill University's Gault Reserve at Mont Saint-Hilaire. Jon Gelhaus and Chris Borkent confirmed identifications of Tipulidae and Mycetophilidae, respectively.

### Funding

This research was supported by a Natural Sciences and Engineering Research Council of Canada Discovery Grant to Terry Wheeler. The funders had no role in study design, data collection and analysis, decision to publish, or preparation of the manuscript.

### Grant Disclosure

The following grant information was disclosed by the authors:
Natural Sciences and Engineering Research Council of Canada Discovery Grant to Terry Wheeler.

### Competing Interests

The authors declare that they have no competing interests.

### Author Contributions

- Julia J. Mlynarek conceived and designed the experiments, performed the experiments, analyzed the data, contributed reagents/materials/analysis tools, authored or reviewed drafts of the paper, approved the final draft.
- Amélie Grégoire Taillefer analyzed the data, contributed reagents/materials/analysis tools, prepared figures and/or tables, authored or reviewed drafts of the paper, approved the final draft.
- Terry A. Wheeler conceived and designed the experiments, performed the experiments, analyzed the data, contributed reagents/materials/analysis tools, authored or reviewed drafts of the paper.

## Data Availability

The raw measurements are provided in Table S1. The data show all the Diptera species that have been identified in this study from fallen logs at different decay stages.

## Supplemental Information

Supplemental information for this article can be found online at http://dx.doi.org/10.7717/peerj.6027#supplemental-information.

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
