# Peer review of "Saproxylic Diptera assemblages in a temperate deciduous forest: implications for community assembly"

_PeerJ, doi:10.7717/peerj.6027_

## Round 0.1 · original submission · Minor Revisions

Two expert reviewers have now assessed your manuscript and both have offered generally favorable comments with suggestions for minor revisions.

I agree with the reviewer assessments and would add one additional minor comment to consider. Specifically, I agree with your interpretation that stochastic influences are likely greater in early decay communities given the higher levels of beta diversity indicated in the NMDS. However, inferring community assembly processes from diversity measures has a history of being controversial so it would be a good idea to add a short discussion that explains why environmental filters might be less important for the communities in early decay stages. For example, these communities could be open to an assortment of generalists leading to early communities that are affected more heavily by dispersal factors, priority effects, and taxonomic drift but that are eventually impacted by increasing colonization of specialists that are potentially superior competitors as more time passes to allow these groups to find the logs. I see this as a simple addition of a few sentences or a short paragraph in the discussion bolstered by relevant references that add a bit more ecological context to how these processes would be at play in this specific system. I know this might seem redundant, but it is a topic worth considering and fleshing out in a bit of detail given the heated debates around community assembly. Given that decaying logs are spatially discrete, you might find some value in considering Jonathan Chase's 2007 PNAS paper on invertebrate communities of small ponds in this interpretation:

http://www.pnas.org/content/104/44/17430

Chase, J. M. (2007). Drought mediates the importance of stochastic community assembly. Proceedings of the National Academy of Sciences, 104(44), 17430-17434.

Congratulations on a nice study. I look forward to your revision.

Reviewer 1 ·

Basic reporting

The paper is mostly fine, but needs a few adjustments to the text:
L53: It's unecessary to list all trophic modes occurring in dead wood, I suggest to delete this parenthesis.
L55: Speight 1989 and Grove 2002 are sufficient references for the definition of saproxylics. The rest can be deleted, no reason to overdo it for such a basic statement.
L101: change from "American beech bark has smoother bark" to "American beech has smoother bark".
L258-263: The meaning of this section seems unclear to me. First of all, there are several Coleoptera associated with specific tree species, as there are references for in the Stokland and Siitonen (2012) chapter cited here. An example is Jonsell et al 1998 in Biodiversity and Conservation. Furthermore, even though the cited papers (Irmler, Rotheray, Persson) might present results in contrast to those of this manuscript, it is unclear why the following sentence stating that few Diptera feed directly on wood would explain that tree species might have a greater influence on Diptera, as opposed to on the many Coleoptera that do feed on wood. One would rather expect the opposite.
L273: The authors state that there were four years between young and old logs, but in the methods section they say decay class of logs was determined by visual inspection. How could the authors know the exact age of the logs?
L293-L295: This sentence is poorly connected with the rest of the section. Also, starting with "even though" one expects the two parts of the sentence to contradict or at least contrast each other, but they rather support each other, so "even though" should be deleted.
L296-L298: This sentence is also poorly connected with the rest of the section. It is also a sentence describing the findings of this paper, which should be clearly stated. Furthermore, I don't understand what is meant by spatial segregation in this context, nor what results are supporting this claim. And I do not see what results support the claim that deterministic community assembly for Dipterans in late decay stages is any less driven by tree species than in earlier decay stages.
L308: Again, what is meant by spatial segregation among species in old logs, and what data supports this statement?
L320-328: This entire section does not belong in the conclusion. It could be moved to another part of the discussion or to the introduction. L311-315 could rather be moved to the conclusion.
I couldn't find the full reference for Persson et al. 2013 in the reference list.

Experimental design

No comment

Validity of the findings

Conclusion is clearly stated in the abstract, but not in the conclusion. See comment under basic reporting.

Reviewer 2 ·

Basic reporting

The article is overall written in professional English and all statements are clearly and unambiguously expressed. There are but a few issues that I want to point out:

Abstract – Line 1: This definition of saproxylic insects is a bit too narrow. Why not use the definition that is given in Line 53-54? The word limit of the abstract should allow this.
Line 44-48: The transition from the 1st to the 2nd paragraph is a bit sudden. There should be at least another sentence on why the authors are interested in the communities of forest dwelling invertebrates.
Line 65: “diversity patterns” would be better and shorter than “patterns of diversity”.
Line 77: The mention of response to disturbances can be omitted since this is not what is examinined in this article
Line 87: “Some” should be replaced with another word, like “clear” or “distinct”. Since this is where the hypotheses are stated, the expectations on stochastic and deterministic processes should also be addressed.
Line 101: This “bark” is redundant – there is another in the next line.
Line 148: Replace “were” with “was”.
Line 158: Insert “a” at the beginning of the sentence.
Line 215: Insert “the” in front of “GLM”.
Line 233: Even though “dipterans” is correct, it should maybe replaced with “Diptera” which is used in all other instances in the text.
Line 269: The author Persson does not need to be named twice within two sentences. Just write “he”.
Line 270: Maybe replace “other” with similar since this study is not focused on Coleoptera or midges
Line 317-328: I would probably change the entire conclusion section. In the current state it almost exclusively discusses stochastic and deterministic processes. This part should rather be put in the section of the discussion where this point is first brought up. The conclusion could instead contain something like the consideration on conservation measures that were put at the beginning of the discussion.

An appropriate number of literature references is used to give necessary background and to discuss the results in light of previous findings and they are all implemented correctly into the text. There is, however, the case of the authors “Persson (2013)” and “Diamond (1975)” who are cited in the text but do not appear in the reference list.
It also seems as if the by PeerJ required citation format has been switched between the text and the reference list: In the list all names should be stated without “&”, while in the text “et al.” should only be applied if there four or more authors – if there are less all are to named with an “&” before the last. This seems not to be case in all citations in the text.

The article is professionally structured in accordance with the requirements of PeerJ. The figures and tables effectively display the results of the study and are correctly labeled and cited in the text. However, Figure 1 would likely look better if the y-axis would at least extend the same level as the highest bars.
I would also maybe do something different with Table 3. An entire table that only shows why the best fitting model was used is not particularly interesting. Instead, the table should be used to describe the model itself – thereby showcasing how the predictors influence the response variable.
Furthermore, it seems to me the supplementary map of the research area has a rather low resolution.
The raw data has been made available clearly structured in an appropriate format.

All in all, the article is an appropriate unit of publication and contains only the results that are relevant to the hypotheses that were posed in its introduction.

Experimental design

The article describes original research in the fields of ecology and zoology and is thereby fully within the aims and scope of PeerJ. Within these fields it poses a relevant research question: Studies on saproxylic invertebrates focus indeed heavily on beetles and therefore this study on Diptera that focuses not on taxonomy, but instead on the mechanism which from their communities, fills a gap in current knowledge. Furthermore, the attempt to discern between stochastic and deterministic processes in communities is not made often and constitutes an interesting approach in this field.

The field experiments have been conducted rigorously and adhere to the technical and ethical standards in this field. A higher number of replicates may have led to more robust results but the sampling effort can still be deemed sufficient for the posed research questions.
All methods have been described in adequate detail to make the experiment replicable at any time.

Validity of the findings

The data is in my opinion robust enough for the statistical analyses that were performed in this study. These analyses are also an appropriate way to test the acquired data with regard to the posed research questions.
In terms of the results it is a bit curious that the individual-based rarefaction curves are that steep in a temperate habitat. However, Diptera are known to be a very species rich group. It is also surprising that only the GLM of abundance lead to significant results, but this might be due to the limited number of replicates.
The conclusions drawn from these results seem sound to me and are competently discussed and compared to similar experiments citing a variety of other authors. At the same time, the limitations of this study are discussed and suggestions for future experiments as well as diversity conservation measures are made. All these conclusions and suggestion are clearly discernible as such.

Additional comments

Overall this manuscript describes an interesting approach and furthers the knowledge on saproxylic communities outside of beetles. The investigation on the importance of stochastic and deterministic processes also adds considerably to this.
With a few minor changes – as suggested above – I would very much approve of this study being published.

Reviewer 3 ·

Basic reporting

See comments

Experimental design

See comments

Validity of the findings

See comments

Additional comments

I found this to be a useful paper, with only a couple of serious problems:



Major points:



l 35 - I don’t think that this opening paragraph is a particularly useful characterization of this debate, nor do I think that this issue is addressed at all in this paper. The debate it is not deterministic vs. stochastic per se, but rather focusses on to what extent individuals of different species can be thought of as more or less identical (e.g., lack of competition or lack of differences in dispersal abilities, etc.). Even those that firmly ascribe to the importance of competition (and niche differentiation) in structuring communities would never deny the importance of stochastic processes (think of islands for example). This paragraph needs to be reworked. Perhaps most importantly, neutral theory is meant to be applied within a guild (trophically similar species), which is not the case here (the various dipterans show huge trophic variation).



line 317 - Similarly, I find the whole deterministic and stochastic aspect in the conclusion to be highly speculative; it does not deserve to be front and centre in your conclusion. You don’t actually have any evidence one way or the other. Instead, focus the conclusion on your actual contributions to the literature (log decay and species).



l 55 - "There is evidence based…." - this sentence needs to be expanded to a full paragraph given the importance of this literature for this paper (there are many more studies to be mentioned). Indeed, this literature should be the focus of the introduction rather than "stochastic" vs. "deterministic" processes! The latter is interesting, but the authors have not collected data in a way that can be used to provide useful tests of the neutral theory.



This paper would be strengthened by downplaying this stochastic-deterministic aspect. I found the discussion on deterministic vs. stochastic processes to be interesting enough to include, but this aspect should be more-or-less removed from the introduction and conclusion. Stay with your strengths (log decay class and species).



l 105 - How were these log ages (approximately 2 and 6 years approximately) determined? From my experience, they appear to be very inaccurate. I was just in a maple stand logged 3 years ago and the bark was intact, and often still firmly attached to the downed wood (not 20% gone). From my experience, the bark will not be 80% absent at six years (it will take considerably longer). Also, what is meant by "the heartwood had been infiltrated with moisture and decay"? How was this determined? This raises a key point: a key objective of this paper is to investigate different decay stages, hence we need accurate and reliable determination of the decay stages (i.e., more than the vague and possibly inaccurate age information provided here). I suggest that the author characterize the logs according to one or more of the "decay class" systems found in the literature (which tend to focus primarily on wood firmness) and provide ages based on empirical information (either their own, or from the literature). At the moment, all we have is bark information, which does little to help use distinguish among latter decay stages (for example, decay stages 3-5 all are characterized by little bark). You return to these ages on line 272, indicating their importance. You need to support them empirically. I do not believe that an estimate of 6 years for the older decay stage is even close to correct (80% bark gone).



More minor points



l 19 - this is not the standard definition of saproxylic, which is more along the lines of "species of invertebrate that are dependent, during some part of their life cycle, upon the dead or dying wood of moribund or dead trees (standing or fallen), or upon wood-inhabiting fungi, or upon the presence of other saproxylics". (SPEIGHT, M.C.D. (1989). — Saproxylic invertebrates and their conservation. Nature and Environment Series, No. 42. Council of Europe, Strasbourg.). You actually have the right definition in the sentence starting on line 51; use it here also.



1 27 - indicators of what? Be explicit.



l 28 - "Both stochastic and deterministic…." Delete the sentence - it is devoid of information (what other kinds of processes are there?).



l 32 - add Diptera to the key words.



l 48 - to characterize bark or wood inhabiting species as having a "significant role in wood decomposition" again is a mischaracterization I think (termites?). See my comment above about the definition of saproxylic as well. Saproxylic organisms do many things and many are reliant on wood decomposition, but whether they have a "signficant role" is more up to debate (for example, do they influence the abundances of white and brown rot fungi?).



l 94 - It is standard to leave a space between numbers and SI units (i.e., use 500 n, not 500m).



l 95 - this sentence implies that old growth forests are the most common type in southern Quebec, which is not true (maple forests, on the other hand, may be common). Clarify.



l 98 - had beech bark disease begun to affect beech trees in the study area at the time of sampling? This may affect subsequent rot processes.



Figure legends - I do not find these sufficiently detailed. Some basic information about the study should be included. The idea is that if someone found the figure and legend on the floor, they would be able to understand what it is about. This can be resolved by adding a sentence or two. Also, in Fig. 1, the light blue looks almost white. Use a better colour (or a pattern).



In Fig. 1, you need to indicate what YM, OM, etc. are. Also, add ± 1 standard error of the mean abundance (across the 5 logs per treatment).



In Fig. 3, it would be better to use standard errors of the mean, so that the statistical significance can be better judged. You note that they are not significantly different, but this is hard to access with just standard deviations.



Please add the actual sample points to Fig. 4. The polygons are good, but it is better with the actual points as well.



line 207 - "no tendency". Why not statistically test this? Did indicator values differ significantly among the trophic groups? (P value?)



line 209 - Why are the observations reported in this paragraph not tested statistically? You have five replicates per treatment after all, and hence can do statistical tests (log-based abundance, richness, diversity, Simpsons). The central limit theorem applies, and hence you may be able to do ANOVAs (or non-parametric if need be). You list AIC values, but have no indication of significance. Are any of the AICs meaningful?



line 215 - you note a "combined effect" (interaction). But, what was the effect? Interaction indicates that the effect of decay class differed among the species. How so exactly?



line 232 - use "relatively distinct" (the polygons show much overlap, and it was only 22% explained).

---

## Round 0.2 · accepted · Accept

I appreciate your efforts to improve the manuscript in light of the comprehensive set of reviewer suggestions and feel at this point you have adequately addressed the limited set of reviewer critiques from the initial review. Please note that PeerJ is a rapid turnaround journal and offers minimal copy editing so please be sure to thoroughly check your forthcoming proofs for grammatical issues and typos.

#